# In-Clinic Measurements of Vascular Risk and Brain Activity

**Jeffrey Boone** [1,2,*], **Anna H. Davids** [3,4], **David Joffe** [5], **Francesca Arese Lucini** [5], **David S. Oakley** [5], **Madeleine J. Oakley** [6,7] **and Matthew Peterson** [8]

1   School of Medicine, University of Colorado, Denver, CO 80045, USA
2   Boone Heart Institute, Denver, CO 80111, USA
3   Eastern Virginia Medical School, Norfolk, VA 23507, USA
4   Saint Joseph Hospital, SCL Health, Denver, CO 80218, USA
5   WAVi Research, Boulder, CO 80101, USA
6   Department of Linguistics, Georgetown University, Washington, DC 20057, USA
7   Department of Linguistics, North Carolina State University, Raleigh, NC 27695, USA
8   Baylor College of Medicine, Houston, TX 77030, USA
*   Correspondence: jboonemd@bonneheart.com

**Abstract:** Background: Cardiovascular disease and dementia represent two health problems that may be causally connected. Studies have shown patients with dementia to have reduced cardiovascular health measures, where patients with dementia also have reduced electrophysiological brain activity as measured by event-related potentials (ERP's). Few studies have attempted to correlate the two: cardiovascular health and ERP brain activity. The objective of this study is to determine if there are ERP differences between patients with lower versus higher measures of cardiovascular risk. Methods: For 180 patients ages 53 (16) years, Audio P300 ERP amplitudes and latencies (speeds) were measured upon initial patient visit alongside other clinical evaluations. Cardiovascular risk was categorized into good versus poor levels for blood pressure resting and stressed, E/A Ratio, atherosclerosis, and carotid intima-media thickness. Results: Groups with good levels had lower latencies (faster P300's) and higher amplitudes than those with poor levels across all cardiovascular risk measures, significant to $p < 0.05$ for most parameters. While both cardiovascular health and P300 metrics decline with age, poor blood pressure and plaque was seen to affect P300 performance across all age groups in this study. Conclusion: These data suggest correlation between brain activity, as measured by the P300, and five standard measures of cardiovascular health and this correlation may begin at an early age. While further explorations are warranted, these results could have implications on the management of preventative medicine by bringing preventative cardiology and brain health together.

**Keywords:** electroencephalogram (EEG); P300; event related potential (ERP); brainwave; cardiovascular disease (CVD)

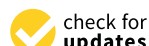


## 1. Introduction

Cardiovascular Disease (CVD) and dementia represent two pressing health problems and there is strong evidence that links the risk factors of both [1–4]. Countries with growing economies are seeing an extreme increase in people developing CVD, already a major cause of death and one of the most frequent causes of hospitalization in the United States [5]. Hypertension (HTN), defined as elevated systolic blood pressure or diastolic blood pressure, remains a leading cause of CVD and the effects of HTN on the brain from CVD are numerous. A recent study of 22,000 healthy individuals, for example, found that elevated blood pressure correlated with reduced executive function [6]. The cognitive decline associated with HTN is the result of vascular changes resulting from long-term, elevated blood pressures. Elevated blood pressure is a direct cause of vascular remodeling and has been repeatedly shown to cause an increase in arterial wall thickness and a decrease in lumen diameter [7,8]. This remodeling in cerebral arteries occurs as an adaptive

process that protects the brain from increased blood pressures. Failure to remodel cerebral arteries can lead to vasogenic edema and a breakdown of the blood–brain barrier resulting in damage to the surrounding cerebral tissues [7]. Likewise, the vascular remodeling undergone to protect the brain from damage as a result of these elevated pressures can have detrimental effects on cognitive functioning as findings such as increased arterial wall thickness and decreased lumen diameter result in decreased blood flow to the cerebral tissues. A recent study found that although the brain continues to create new neurons throughout old age, angiogenesis, the formation of new blood vessels, slows with age causing a decrease in cerebral blood flow and a resultant decrease in neuroplasticity [9]. The study's findings suggest that through behavior modification to increase cerebral blood flow, it may be possible to decrease the risk for age-related cognitive decline.

Dementing diseases affect 5.6 million Americans, and the associated costs are approaching $290 billion annually [10,11]. A recent Harris poll confirms the generally accepted idea that populations of all ages, and particularly the baby boomer population, is increasingly concerned about mental decline in the aging process, with those over 50 citing Alzheimer's Disease as their greatest fear even over cancer and heart disease [12]. Dementing diseases notwithstanding, as the population ages, a wealth of cultural value is lost as general mental performance declines—declines that are a part of normal aging and not dementia.

While cardiovascular risk factor modification may play an important role in brain health and cardiovascular health measurements are commonly used in clinical practice, physiologic measurements of brain activity are virtually absent in clinics except for patients with a pending neurologic diagnosis. Given the advent of new technologies, in-clinic measures of brain function can now be obtained alongside standard cardiovascular health measures and the goal of this study, therefore, is to investigate the relation between commonly measured cardiovascular-risk markers and brain activity to identify potential correlates. These are important questions where the hope is that risk factor modification may prove as effective in the prevention of dementia as it has for CVD [13]. In addition, given the climate of fear surrounding Alzheimer's Disease and other dementias, this brain-heart information may also aid in patient compliance to prescribed interventions.

*1.1. Clinical Measures*

1.1.1. Vascular Risk

This study focuses on five common clinical measures of vascular risk: blood pressure during stress (BPS), blood pressure resting (BPR), presence of atherosclerosis (Plaque) in the common carotid artery, carotid intima-media thickness (CIMT), and E/A Ratio.

Blood pressure is the most common clinical measure of cardiovascular health where decreasing blood pressure levels have been associated with a decreased risk of cardiovascular events [14,15]. As discussed, HTN is a major health concern that is pervasive across class, gender, and race [16,17].

Plaque is another commonly measured clinical marker as atherosclerosis has long-been thought to be a leading cause of CVD and stroke, where plaque development can lead to a greater arterial wall thickness, atherosclerosis, and acute coronary syndromes [18,19]. Atherosclerosis is a buildup of plaque in the arteries caused from excessive amounts of cholesterol and other products [20]. Presence of plaque can be an important cardiovascular risk factor and early detection is often considered a top clinical priority [21]. CIMT, an ultrasonic measure of carotid thickness, can also provide an affordable and noninvasive option for monitoring the development of atherosclerosis and is therefore included in this study [20,22].

Finally, E/A ratio is another common measure included in this study. Diastolic heart failure, or heart failure with preserved ejection fraction, is cited by the American Academy of Family Physicians as a major cause of morbidity and mortality in patients with cardiovascular risk factors [22]. The most common cause of diastolic heart failure is chronic, uncontrolled HTN which leads to left ventricular hypertrophy and decreased compliance [23]. It is additionally more common in the elderly as aging leads to increased collagen

cross-linking and loss of elastic fibers which causes ventricular stiffness and decreased compliance [24]. Diastolic dysfunction is an early sign of diastolic heart failure and Doppler echocardiography has been used as a noninvasive tool for detecting abnormalities by measuring the ratio between the peak velocity of early filling (E) and the peak velocity of late filling (A) [22,25]. Monitoring E/A Ratio may assist in early identification of diastolic abnormalities [26].

### 1.1.2. Brain Activity Measures

This study utilizes event-related potentials (ERP) as an in-clinic measure of cerebral activity. ERPs are a measurement of the electroencephalogram (EEG) signal time-locked to the onset of a given stimulus and consist of different components labeled by their polarity (P for positive or N for negative) and their time of occurrence after the stimulus in milliseconds (e.g., P300).

This study employs the P300 during an audio oddball task. This protocol is well suited for in-clinic measurements because it has been extensively studied, is readily standardizable and can be implemented on a large scale [27]. The P300 parameters reported here include amplitude and latency that focus on the brain's recognition of the odd tone as different as measured by cortical voltage changes, most reliably in the central-parietal regions. P300 amplitude is thought to be proportional to the number of attentional resources devoted to a given task where P300 latency (the delay between stimulus delivery and recognition of the oddball tone as different) is a measure of stimulus classification speed [28]. As an increase in the P300 peak latency, and/or a decrease in the P300 peak amplitude are observed in various conditions accompanied by the impairment of cognitive functions, including aging, dementia, mild traumatic brain injury (mTBI), and depressive disorders, these measurements are considered nonspecific [29].

While P300 has been investigated for a large variety of conditions, few studies have focused on heart health. One study found delays in P300 latencies in hypertensive elderly patients, relative to normotensives but the sample size was not large enough to draw any statistically significant conclusions regarding P300 and blood pressure [30].

The main objective for this study is to use an audio P300 protocol alongside the above-mentioned measures of heart health to determine if there is a correlation with heart health and cognitive brain processing. Specifically, is there a statistically significant difference in P300 parameters between those with healthy and unhealthy cardiovascular biomarker levels?

## 2. Materials and Methods

### 2.1. Subjects and Clinical Procedure

The subjects for this study were comprised of all healthy subjects who visited the Boone Heart Institute for a combined preventative cardiology and EEG/ERP evaluation for the 2-year duration of the study. This clinic is self pay and its patients enroll in the program for health optimization and not to treat a disease state. Exclusion criteria included taking beta-blockers or psychiatric medication, had a history of stroke, or suspected dementia, and those who had lower than 80% yield on the audio P300 protocol due to artifact. Out of the total sample size of 321 subjects, heart-health and brain-function parameters were obtained on 180, of whom 119 were male and 61 female and with an average age of 53 (16) years. Of these 180 patients, 21 were excluded based on the criteria outlined above. The study was approved by the Solutions Institutional Review Board and written informed consent was obtained from the participants before study intake.

The evaluation included a blood test, WAVi Brain Assessment which included EEG, and audio P300, blood pressure readings at rest (BPR) and during stress on the treadmill (BPS), genetic tests, carotid ultrasound, Doppler echocardiography, and electrocardiogram. Measures relevant for this study are audio P300, blood pressure (sitting and stress), E/A Ratio, and CIMT using standard B-mode ultrasound with measures obtained using multiple

carotid images one centimeter proximal to the carotid bifurcation and extracted with Sonocalc software (SonoSite Inc., Bothell, WA, USA) (https://www.sonosite.com/).

After fasting overnight, subjects arrived at Boone Heart Institute in the morning (9 a.m.–12 p.m.) for a blood test, blood pressure readings, treadmill and other standard clinical evaluations. Immediately following, subjects were allowed a small snack (i.e., Trail Mix), hydration, and time to relax before the WAVi Brain Assessment.

### 2.2. EEG Acquisition and Preprocessing

The EEG was recorded using the WAVi® Research Platform (WAVi Research, Boulder, CO, USA) sampled at 250 Hz and bandpass filtered between 0.5–30 Hz. The electrodes were placed according to the International 10–20 system using caps with 19 tin electrodes (both with the WAVi Headset and Electro-cap International Inc., Eaton, OH, USA). Linked reference electrodes were placed at the earlobes.

The test administrators were instructed to establish electrode impedances below 30 kΩ for EEG locations and below 20 kΩ for the ground-to-ear locations where possible. These targets are well below the 1 GΩ input impedance of the WAVi amplifiers, are practical regarding preparation time, and produced sufficient yield [31].

To be consistent with the goal of administering a simplified test, a continuous 4 min 2-tone audio oddball eyes- closed P300 protocol was used. Here, 200 common tones (1000 Hz) and 40 rare tones (2777 Hz) were delivered in random order over the span of 4 min, using a 0.95 s interstimulus interval with a 50 ms tone length. The tones were delivered using Skullcandy™ over the ear headphones, at 65 dB. The time, and strength, of the brain's identification of the rare tone provides the brain metrics for this study.

### 2.3. EEG Extraction

The WAVi Research Desktop V 0.9.7.2 was used to extract and analyze the EEG data, which included automatic artifact (noise) rejection where EEG segments with excessive amplitude and/or frequency activities were automatically excluded from analysis on a channel-by-channel basis. While many methods exist for removing movement and other artifacts, this technique produces acceptable test–retest variance. Data were then visually inspected for noise to verify the accuracy of the automated artifacting.

P300 components were measured by identifying the positive extremum in the latency range of 240–500 ms. The depth (P300V) was then extracted from the mean amplitude of all stimuli and the latency (P300T) is the delay recorded for that depth. Each independent ERP epoch was baseline corrected using the 100 ms pre-stimulus period.

P300 parameters are typically extracted from the Cz, or Pz, or the average of various sites. Here, we report a P300 μV that is the highest amplitude from the 6 central-parietal (C-P) scalp sites (C3, Cz, C4, P3, Pz, P4), and the fastest P300 time (smallest latency) from these same 6 C-P sites. These sites both produce an acceptable test–retest variance but have also been noted as the most useful for mTBI identification in previous studies [32,33].

### 2.4. Statistics

The goal of this paper is to compare ERP values between groups of healthy individuals deemed to have good cardiovascular markers and those deemed poor according to the Boone Heart Institute protocol (Table 1). As this clinic focusses on health optimization or prevention and not on disease states, these procedures may be more stringent than other standards, though the values of good and poor blood pressure targets are consistent with previously cited studies [3,6]. Note that in the case of blood pressure and E/A ratio, the middle range has been excluded in order to create the two analysis categories. In Table 1, either poor systolic OR poor diastolic blood pressure qualifies as poor blood pressure for this study.

**Table 1.** Good vs. Poor Levels for the heart biomarkers used in the study, with ranges following the protocols of the Boone Heart Institute.

| Heart Biomarkers | Good Levels | Poor Levels |
|---|---|---|
| Blood Pressure (stress) | <130/90 mmHg | >141/101 mmHg |
| Blood Pressure (rest) | <120/80 mmHg | >136/91 mmHg |
| Plaque | No | Yes |
| CIMT | <0.6 mm | >0.61 mm |
| E/A Ratio (supine) | >1.3 | <1.1 |

The groups of Table 1 were compared using unpaired two-sample *t*-tests. As studies have shown that P300 parameters decline with conditions that affect cognition, we hypothesize that P300 metrics should improve when cardiovascular biomarkers are deemed healthy. We therefore use a 1-tailed t-test and set our significance to $p < 0.05$ in comparing group differences. Note that we also compared P300 parameters between medication categories, discussed below. In these cases, we use a 2-tailed test. In response to a concern over the lack of reproducibility of certain medical studies, we set our *p* value cutoff to 0.05 with a minimum meaningful effect size of Cohens D > 0.50 [34].

## 3. Results

P300 responses were larger and faster for all groups with the healthier cardiovascular markers, averaging a 16% increase in voltage and 8% decrease in latency for the healthy groups across all clinical measures (Table 2).

**Table 2.** Differences in P300 between subjects classified as good versus poor for the cardiovascular biomarkers of Table 1. Bold values represent significant differences, $p < 0.05$, and large effect sizes, Cohen's D > 0.55. No effect sizes are given for non-significant parameters.

| Heart Biomarkers | N | P300T (SD) | P300V (SD) |
|---|---|---|---|
| BP Stress Good | 28 | 304 (44) ms | 16.1 (7.7) uV |
| BP Stress Poor | 80 | 325 (51) ms | 11.9 (4.6) uV |
| *p* Value (CohD) | | 0.020 (0.44) | 0.005 (0.60) |
| BP Rest Good | 30 | 306 (47) ms | 16.0 (7.8) uV |
| BP Rest Poor | 58 | 322 (50) ms | 11.3 (4.3) uV |
| *p* Value (CohD) | | 0.061 | 0.002 (0.79) |
| EA Ratio Good | 50 | 302 (43) ms | 13.6 (5.7) uV |
| EA Ratio Poor | 69 | 327 (63) | 12.7 (5.5) uV |
| *p* Value (CohD) | | 0.005 (0.46) | 0.2 |
| Plaque Good | 46 | 293 (35) ms | 14.2 (5.7) uV |
| Plaque Poor | 105 | 327 (53) ms | 12.5 (5.7) uV |
| *p* Value (CohD) | | <0.001 (0.76) | 0.047 (0.33) |
| CIMT Good | 84 | 308 (46) ms | 13.0 (6.1) uV |
| CIMT Poor | 68 | 327 (54) ms | 13.0 (5.2) uV |
| *p* Value (CohD) | | 0.014 (0.39) | 0.3 |

Figure 1 shows the result for patients with plaque and patients without plaque, both as topographs highlighting voltage amplitudes across the cortex and as voltage versus time plots. Figure 2 shows voltage plots for blood pressure groups. In both plots the delay and reduction of the P300 evoked response is evident (large positive downward peak near

300 ms) for the groups with poorer heart health. It is interesting to note that this reduction is not seen on the frontal locations and only at the central-parietal locations where the P300 is strongest and thought to be most reliable.

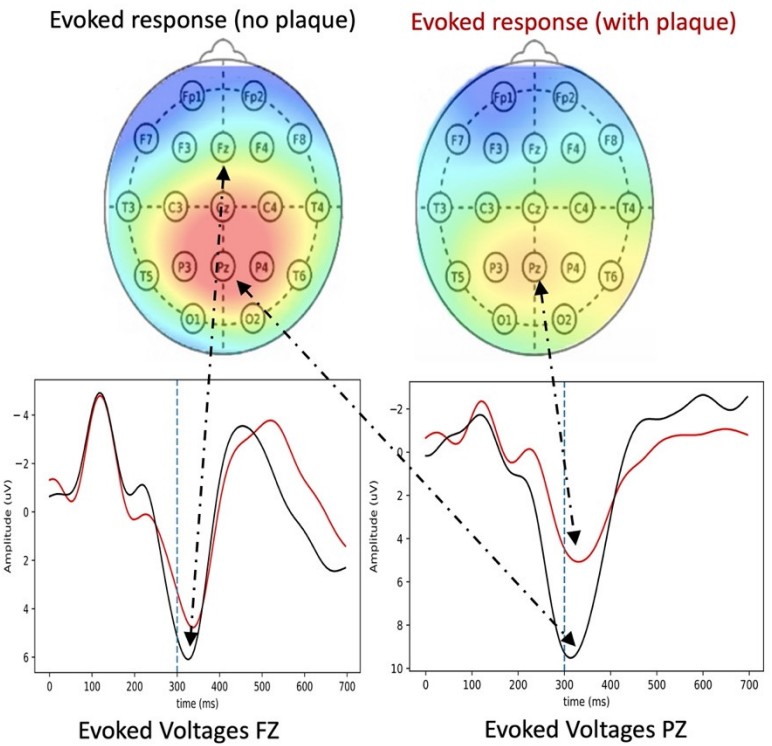

**Figure 1.** P300 response for patients with and without plaque. Top display shows P300 amplitudes (blue–red = 0–10 uV) across scalp locations and bottom shows response voltages as a function of post-stimulus time for two scalp locations (frontal Fz and parietal Pz). Arrows illustrate the link to scalp locations (bottom plots) where the black curves represent voltages averaged for all patients without plaque and the red curve is the average for all patients with plaque. Note that it takes around 300 ms for the brain to cognitively distinguish the novel signal (vertical blue lines) as indicated by the prominent positive downward peak. This study utilizes the depth and timing of this peak as dependent variables of interest. Here, it can be seen that the patients with plaque had on average a smaller and slower parietal (but not frontal) P300 response than those without.

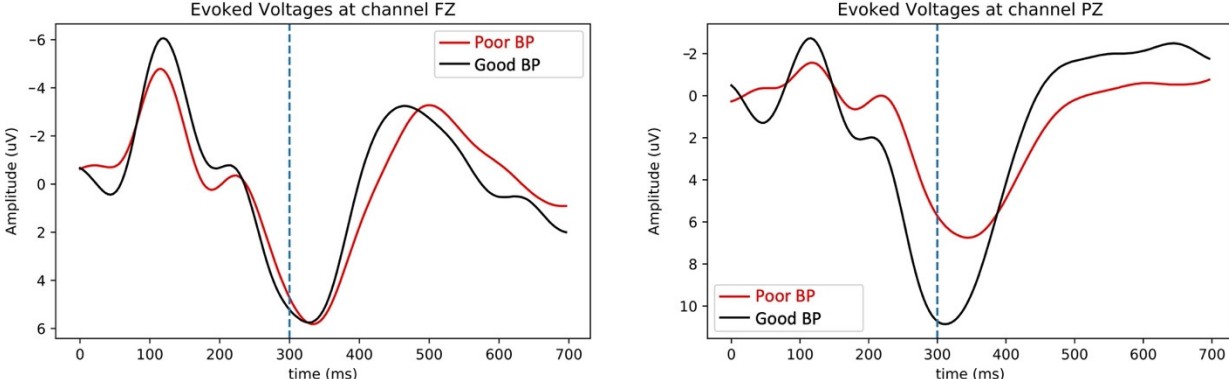

**Figure 2.** Evoked responses for patients with Good and Poor blood pressure (stressed) as a function of post-stimulus time for two scalp locations (frontal Fz scalp location and parietal Pz location). Black curves represent voltages averaged for all patients with good levels and the red curve is the average for all patients with poor levels. Here, it can be seen that the patients with poor blood pressure had on average a smaller and slower parietal P300 response than those with good levels. Again, we do not see this effect in the frontal locations.

The higher P300 amplitudes were statistically significant ($p < 0.05$) for the blood pressure stressed, resting, and E/A groups, whereas Plaque and CIMT differences were not (Table 2). The faster P300 speeds (lower latencies) were significant for all groups with the healthier cardiovascular markers across all clinical measures ($p < 0.05$ with medium to large effect sizes) for all parameters except blood pressure sitting. The blood pressure differences are comparable in direction to a previous study with a smaller sample size [32].

All clinical measures agreed with the hypothesis in that the direction of the effect showed increased P300 parameters (speed and voltage) with better cardiovascular health (as shown in Figures 3 and 4).

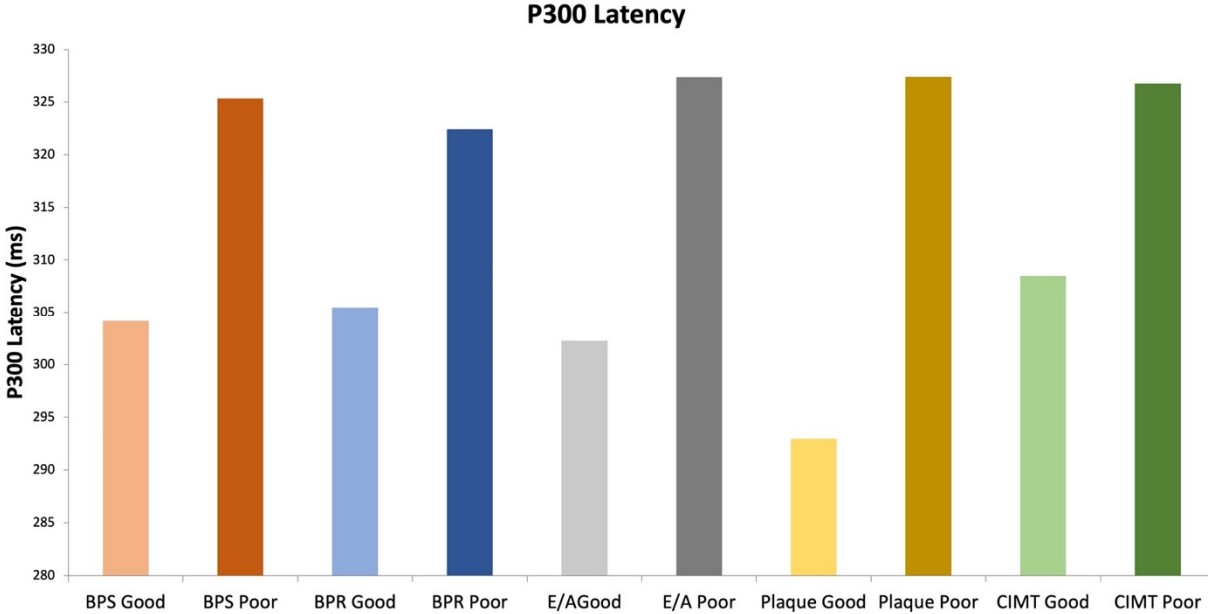

**Figure 3.** The average P300 latency for all tested heart biomarkers comparing Good versus Poor Levels. In all categories, Good Levels displayed faster latencies than for Poor Levels.

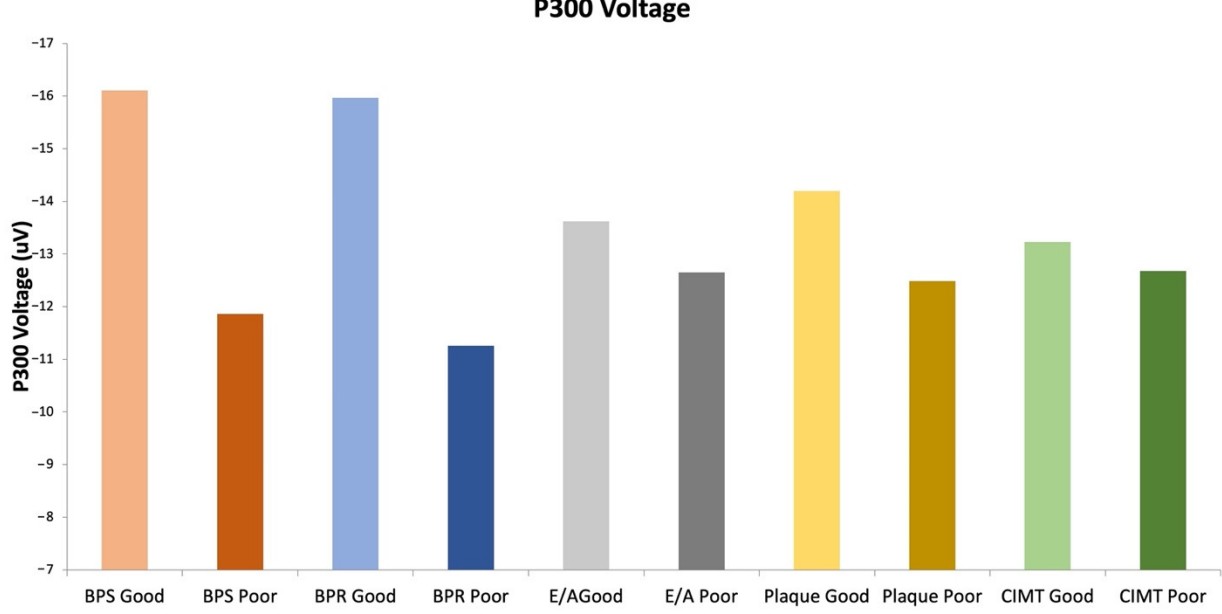

**Figure 4.** The average P300 amplitude for all tested heart biomarkers comparing Good versus Poor Levels. In all categories, Good Levels displayed larger amplitudes than for Poor Levels.

As this is an in-clinic study with limited exclusion criteria, it's important to understand medication as a confounding variable. Since this population is prescribed a variety of medication, in order to test possible medication affects we divided the medication groups into 5 subgroups: Statins, Blood Pressure, Psychiatric, Other, and None. Table 3 shows the effects of the various medications for this population compared to the entire data set. The only significant finding was a decrease in latency (increase in speed) for the excluded psychiatric group. While we use a simple t-test here because we are interested in exploring differences from the study population, a single set, these findings were also confirmed by ANCOVA tests, which is used to test the effects of categorical variables (medications) on a continuous dependent variable (P300V/P300T), controlling for the effects of selected other continuous variables (age), which vary with the dependent (P300V/P300T). Both P300 Voltage and P300 Delay depend on age (quantitative covariate) and medications (single-factor) and the main question we want to ask when doing ANCOVA is, once we account of changes in P300V/P300T due to the age of the patients, what is the effect on P300V/P300T that is left due to the medications that the patient is taking? The ANCOVA analysis showed that the variation, accounted for by age alone, and then by medication, are not significant, in any of the two cases. This is due to the fact that there is more variance in P300 voltage and time within each medication group (illustrated in Table 3) than between medication groups. While the sample size does not warrant making any inferences from the statin group, this potential voltage *increase* should be explored further.

**Table 3.** Differences in P300 between subjects for each medication group as compared to the entire study sample. "Other" refers to those not on statins, psychiatric medications, or blood pressure medication. A 2-tailed *t*-test was utilized. Bold values represent significant differences, $p < 0.05$, and large effect sizes (D) Cohen's D > 0.55. No effect sizes are given for non-significant parameters.

| Medication | P300T (SD) ms | *p* Value (D) | P300V (SD) uV | *p* Value (D) | N |
|---|---|---|---|---|---|
| Population | 317 (50) | | 13.0 (5.7) | | 152 |
| Statin | 316 (49) | 0.9 | 14.6 (7.2) | 0.2 | 23 |
| Psychiatric | 275 (25) | **0.03 (1.0)** | 14.7 (7.4) | 0.5 | 7 |
| Blood Pressure | 313 (22) | 0.7 | 12.4 (5.2) | 0.7 | 16 |
| Other | 310 (100) | 0.7 | 13.0 (7.0) | 0.9 | 20 |
| None | 313 (77) | 0.7 | 13.3 (4.9) | 0.8 | 30 |

*Age Trends*

Blood pressure and plaque concerns can begin early, often before the age of 30 [35,36], and this study population also follows this trend (as Figure 5 suggests). Likewise, P300 parameters generally decline with age, where decreases are on the order of 1 uV and 10 ms per decade are reported, with more stability expected in the 35–65 age range [30,37]. It would be interesting to determine the relationship between age-related cognitive decline and cardio health. While answering this question is beyond the scope of this study, we did find that age was not a significant variable (discussed above) and also found that young adults in this study (32 (10) years) with poor blood pressure also have significantly reduced P300 voltage (Table 4). This is illustrated in Figure 6, where the population of all subjects under the age of 55 in this study show similar cardio-related trends seen in Figures 1 and 2 for the full population aged 17–95 years.

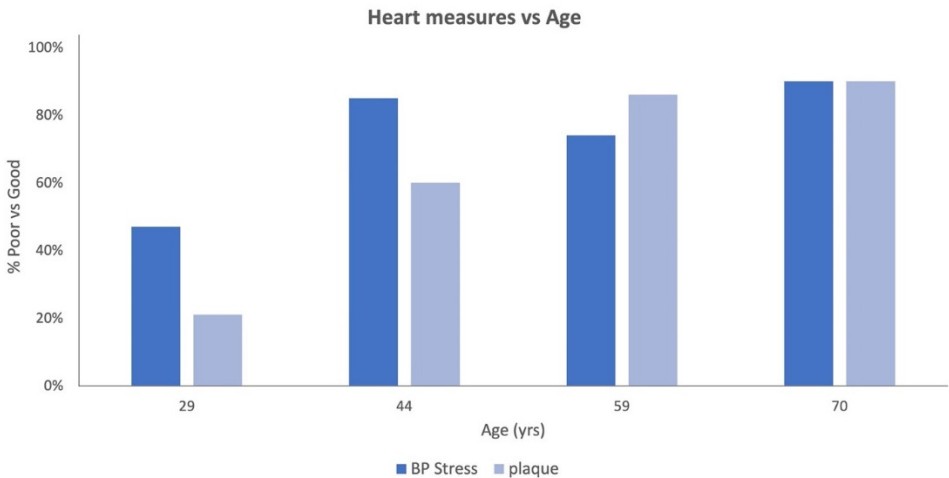

**Figure 5.** The percentage of patients in this study showing Poor BPS vs. Good BPS and Plaque vs. No Plaque (as per Table 1) as a function of age.

**Table 4.** Differences in P300 between younger subjects classified as good versus poor blood pressure measures. Bold values represent significant differences, $p < 0.05$, and large effect sizes, Cohen's D > 0.55.

| Heart Biomarkers | N | Age (SD) | P300T (SD) | P300V (SD) |
|---|---|---|---|---|
| BP Stress Good | 13 | 33 (12) years | 292 (31) ms | 16 (8) uV |
| BP Stress Poor | 10 | 32 (10) years | 307 (46) ms | 11 (6) uV |
| *p* Value (CohD) | | | 0.17 (0.39) | 0.04 (0.79) |

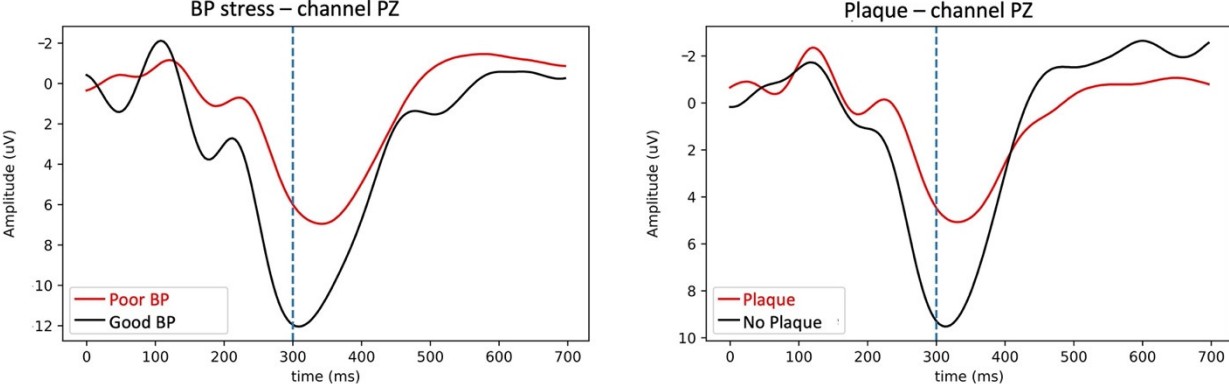

**Figure 6.** Evoked responses at Pz for good and poor levels of blood pressure **(left)** and plaque **(right)** averaged for all patients in this study younger than 55 years. Red curves represent poor levels and black represent good. Note that the trends of Figures 1 and 2 remain where poor cardio health corre-sponds to low P300 performance even for this younger population.

## 4. Discussion

The purpose of the present study was to investigate the relationship between a clinically accessible measure of brain activity, the audio P300 during EEG, and CVD. The P300 is an objective measure of cognitive brain processing that can assist physicians in compliance and tracking. These P300 results were compared between good and poor levels of multiple cardiovascular measures and it was found that faster speeds correlated with healthier heart metrics. Similar results were witnessed when investigating P300 amplitudes where higher voltages were seen in the healthy-heart groups. Significant results were also found to have medium to large values for effect size and support previous studies linking heart health

to brain health [1–8]. The results of this study may also support a case for more stringent blood pressure targets in order to improve cognition [3].

To investigate additional confounding variables, we examined the effect of medications on P300 values. The only statistically significant finding we observed was a decreased latency with psychiatric medications. While there were insufficient numbers to fully investigate statins, the increased voltage is intriguing and warrants further study, particularly in a test–retest study.

Finally, we sought to investigate age as another confounding variable, where P300 parameters decline with age as do cardiovascular measures. Here, we found significant reduction in P300 voltage with poor blood pressure even in the young adult group ages 30 (10) years. This is consistent with previous work that found an association between high blood pressure and poor executive function for ages 44–69 (but in this study not for the >70 age group) [6]. While more studies are needed, a link between blood pressure and cognitive function at earlier ages could have profound implications for helping reduce age-related cognitive decline.

## 5. Limitations and Future Directions

Caution is needed when retrospectively analyzing data collected in the course of routine clinical evaluations. First, the study samples overlap in that some patients with Good Blood Pressure were in the Poor Plaque group, etc. This may offer some statistical bias. The determination of what constitutes "Good" versus "Poor" levels itself is clinically constructed and open to further discussion. Additionally, the medication and medical history may not be complete, thus adding potential confounding variables which need to be further investigated. Finally, the results of this study are mainly associative in nature, thus future studies should be designed to determine if interventions designed to improve cardiovascular health also improve cognitive health.

Age effects need to be further explored. The number of people in each age group, and each cardio group, was limited to the population visiting the clinic. While the results suggest associations at earlier ages for all heart metrics, only the blood pressure correlations were significant. Associations between high blood pressure in young adults, a treatable condition, and future cognitive decline would be an important finding for dementia risk-factor modification.

Finally, while this dataset is structured in such a manner as to be suitable for other analysis methods, including but not limited to spectral or machine-learning techniques, this analysis only focused on the P300 components in the central-parietal regions. The possible lack of change in the frontal regions is an interesting finding that should be explored further.

## 6. Conclusions

Our results indicate that there is a positive association between markers of cardiovascular health and markers of brain health. In particular, we observed higher P300 amplitudes and shorter P300 latencies in groups with healthier cardiovascular measures. These observations corroborate previous results suggesting that poor cardiovascular health can affect cognition. These results also show that high blood pressure can affect cognitive P300 responses even for young adults, which may prove important for dementia risk-factor modification.

While the causality of these results, including age-relations, warrant further exploration, they bring preventive cardiology and cognitive health together with an objective measure of brain activity that is clinically accessible and can assist physicians tracking and compliance.

**Author Contributions:** Data curation, J.B., D.J. and M.J.O.; Formal analysis, A.H.D., F.A.L. and D.S.O.; Investigation, A.H.D. and M.J.O.; Methodology, A.H.D., M.J.O. and M.P.; Project administration, J.B. and M.P.; Resources, J.B.; Software, D.J.; Writing—review and editing, F.A.L. and D.S.O. All authors have read and agreed to the published version of the manuscript.

**Funding:** This research received no external funding.

**Institutional Review Board Statement:** The study was conducted in accordance with the Declaration of Helsinki, and approved by the Solutions Institutional Review Board.

**Informed Consent Statement:** Informed consent was obtained from all subjects involved in the study.

**Data Availability Statement:** Access to these data can be requested by qualified researchers engaging in independent scientific research and will be provided following review and approval of a research proposal and statistical analysis plan and execution of a Data Sharing Agreement. For more information or to submit a request please contact David Oakley, davido@wavimed.com.

**Conflicts of Interest:** David Joffe, Francesca Arese Lucini, and David Oakley are paid consultants and employees of WAVi Co., the company who loaned the equipment to Boone Heart. No other authors have conflicts to report.

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
