# Peer review of "In-Clinic Measurements of Vascular Risk and Brain Activity"

_2673-9259, doi:10.3390/jal2030020_

Round 1

Reviewer 1 Report

Dear authors,

O have read with interest your paper. Here are my observations:

- In the abstract, you must correct „18p” to 180

- In my opinion, you cannot write a whole „Discussion” section without having any cited references. It is impossible to write discussions without citing proper previously published work and comparing it with your own results.

- the study is fairly conducted, but the lack of proper discussions makes it look like it is finished in a hurry.

Good luck in improving it! 

Best regards,

L.C.

Author Response

We thank the reviewer for helpful suggestions (and one typo) for the manuscript.

  • Good catch on the abstract. Don’t know how this happed but we changed 18p back to 180. Maybe this was a formatting issue discussed in the cover letter.
  • We enhanced the discussion and conclusion sections and added a limitation section. The reviewer is correct and this made for a better paper.

Reviewer 2 Report

In this study, the authors investigated the correlation of vascular risk to brain activity, using the p300 latency and voltage as the readout. The major concern is that the more appropriate statistical method would be applied to support the conclusion, in addition to the t-test performed in the manuscript. Specific comments were provided below:

1.      Table 1: The authors are encouraged to describe how they set up the cutoff values for “good level” and “poor level” in terms of CIMT (0.6mm) and E/A ratio (>1.3 and <1.1). Is it based on previous studies? 

2.     Given that the medication and age have the potential to alter the p300 value, it would be appropriate to consider the medication and age as covariates, and the ANCOVA test should be performed accordingly.

3.     Table 2: the average ±SD of p300T and p300V should be provided for each group. 

4.     Figure 1: the scale of color code should be provided; please check that the label of the arrows is correct and describe what the arrows indicate in the legend. 

5.      Figure 2: the systolic cutoff value (150) is different as it is in Table 1 (<130 and >141). The authors should provide the reasons. This issue also applies to Figure 6. As the curves represent the average values from all the patients, an SD bar should be provided for each data point. This issue also applies to Figure 6.

6.     Please make consistent labels of “ p300 latency (SD)”, “p300 voltage (SD)” in table 3 and table 2 “p300T”, “p300V”. 

Author Response

We thank the reviewer for careful attention to the manuscript.

  • The levels were chosen by the clinic for its preventative health optimization program, which are more stringent than some standards. We added a few sentences clarifying this and cited a few references as well.
  • We ran an ANCOVA and found no differences between the groups. We added a sentence to the manuscript but do not want to confuse the fact that the main goal of the test was not to compare the meds to each other but rather to compare to the group as a whole.
  • We were trying to keep tables short but reviewer is correct in wanting values (SD). We changed the table.
  • We changed the Figure and caption. The typo on the arrows was a good catch by the reviewer.
  • Another good catch by the reviewer. We corrected the typo in the figure legend.
  • Even though it was described in the text, we should be consistent P300T vs latency labels etc. We changed the table.

Reviewer 3 Report

I want to thank the authors and the Editorial Board for the opportunity to review the article submitted to the Journal of Ageing and Longevity. The authors’ manuscript refers to a very important topic, vascular health. I believe that the presented work shows important results from a practical point of view. Therefore, I recommend its publication after a few minor changes.

Please describe if information about patients’ diagnosis and/or NYHA class is available. If not, I believe it should be put as a limitation since those can act as significant covariates of the measured results.

I recommend that authors separate effect sizes from p-values. For example, in Table 2, I believe it would be clearer to put Cohen’s d in a separate column for bigger visibility.

I suggest that authors supplement the results of the t-test analysis with 4 additional statistics: t-value, degrees of freedom, lower bound confidence interval and upper bound confidence interval.

I believe that the manuscript would also benefit from adding the visualisation of SD in Figure 3.

X-axis on Figure 5 does not correspond to its description. Authors write that “individuals younger than 55 years in this study, as a group, also experience the same cardio-correlated cognitive decline…”, but Figure 5 shows different age groups. What is more, based on the authors’ results, they could correlate age (as a continuous variable) with other measured variables. Since age’s M=53 and SD=16, it can be included in correlation analysis due to its variance. Without proper correlation analysis (with the usage of e.g. Pearson’s r test) suggestions in section 3.1 are not supported.

Lastly, I suggest that authors supplement their manuscript with a limitations section.

Author Response

The reviewer was very helpful.

  • The patients are healthy and entering a self-pay optimization program and not for a disease condition. We made note of that in the text because that would indeed add a lot of confounders.
  • We expanded Table II. Tried to keep it concise but reviewer is correct.
  • We are trying to keep the tables concise: N gives information on the degree of freedom and we think that adding t-values would add rows/columns whereas the p-value/effect size tells the story: is there a difference and what is the effect size? We want a simple story for the reader.
  • That made it messy but we did include values in Table II
  • Figure 5: If the reviewer meant to say “the y-axis” did not correspond to the description, the reviewer was very observant and that was a good catch. We corrected the caption. We also cleaned up the language to make it clear that Figure 5 was an illustration of younger people in the study also measured with poor heart levels. We then added a table of a blood pressure correlation to support some of the findings and finally we discussed this again in the limitations section. Whew.
  • We referred to limitations in the text but it reads better with an actual limitation section. Thanks.

Reviewer 4 Report

The authors have investigated the relation between cardiovascular fitness/risk factors and brain activity. The study tries to show how worse CV status in associated with less brain activity. The idea of the study is interesting and is worth publishing. The study is overall easy to read and set to a broad audience. However there are major concerns to be addressed:

- The main problem with the study is how cardiovascular status was defined. The authors chose arbitrary values for group differentiation. It is mandatory to consider the main CV guidelines available, either ESC, JACC or AHA.

-The population demographic is also inadequate. Besides the CV markers used in the study it is also necessary to know the patients diseases and time of disease.

-Future prospectives and limitations are missing  

Author Response

The reviewer had interesting observations.

  • The levels were chosen by the clinic for its preventative health optimization program, which are more stringent than some standards. It is also consistent with findings of previous work. We added a few sentences clarifying this.
  • See above. This is an important point b/c disease states are a cofounder but these patients didn’t enter the program to treat disease states and we made note of that.
  • Reviewer is correct. Though we did discuss limitations these were better in their own section and so we added one.

Round 2

Reviewer 4 Report

The Authors have addressed the critical aspects of the paper. The article is now fit for publication.